# Long-time prediction of nonlinear parametrized dynamical systems by deep learning-based ROMs

**Federico Fatone**
MOX - Department of Mathematics
Politecnico di Milano
federico.fatone@mail.polimi.it

**Stefania Fresca**
MOX - Department of Mathematics
Politecnico di Milano
stefania.fresca@polimi.it

**Andrea Manzoni**
MOX - Department of Mathematics
Politecnico di Milano
andrea1.manzoni@polimi.it

## Abstract

Deep learning-based reduced order models (DL-ROMs) have been recently proposed to overcome common limitations shared by conventional ROMs – built, e.g., through proper orthogonal decomposition (POD) – when applied to nonlinear time-dependent parametrized PDEs. Although extremely efficient at testing time, when evaluating the PDE solution for any new testing-parameter instance, DL-ROMs require an expensive training stage. To avoid this latter, a prior dimensionality reduction through POD, and a multi-fidelity pretraining stage, are introduced, yielding the POD-DL-ROM framework, which allows to solve time-dependent PDEs even faster than in real-time. Equipped with LSTM networks, the resulting POD-LSTM-ROMs better grasp the time evolution of the PDE system, ultimately allowing long-term prediction of complex systems' evolution, with respect to the training window, for unseen input parameter values.

## 1 Introduction

PDEs are extensively used for the mathematical description of several physical phenomena. However, traditional high-fidelity, full order models (FOMs) employed for their numerical approximation, such as those based on the finite element method, become infeasible when dealing with complex systems and multiple input-output responses need to be evaluated (like, e.g., for uncertainty quantification, control and optimization) or real-time performances must be achieved. In this context, projection-based reduced order models (ROMs) – such as POD-Galerkin ROMs – have been introduced with the goal of enhancing efficiency in timing critical applications. Nevertheless, despite being physics-driven, POD-Galerkin ROMs show severe limitations when applied to nonlinear time-dependent PDEs, which might be related to (i) the need to deal with projections onto high dimensional linear approximating trial manifolds, (ii) expensive hyper-reduction strategies, or (iii) the intrinsic difficulty to handle physical complexity with a linear superimposition of modes [14].

To overcome these drawbacks, several nonlinear projection methods have been introduced, the majority of which takes advantage of artificial neural networks [6,7,13,1,12,10]. A recently proposed strategy [3] aims at constructing DL-based ROMs (DL-ROMs) for nonlinear time-dependent parametrized PDEs in a non-intrusive way, approximating the PDE solution manifold by means of a low-dimensional, nonlinear trial manifold, and the nonlinear dynamics of the generalized coordinates on such reduced trial manifold, as a function of the time coordinate and the parameters. The former is learnt by means of the decoder function of a convolutional autoencoder (CAE) neural network; the

35th Conference on Neural Information Processing Systems (NeurIPS 2021), Sydney, Australia.

latter through a (deep) feedforward neural network (DFNN), and the encoder function of the CAE. DL-ROMs outperform the reduced basis method – regarding both numerical accuracy and computational efficiency at testing stage. With the same spirit, POD-DL-ROMs [2] enable a more efficient training stage and the use of much larger FOM dimensions, without affecting network complexity, thanks to a prior dimensionality reduction of FOM snapshots through randomized POD (rPOD), and a multi-fidelity pretraining stage, where different models (exploiting, e.g., coarser discretizations or simplified physical models) can be combined to iteratively initialize network parameters.

This work extends the POD-DL-ROM framework [2] in two directions: first, it replaces the convolutional autoencoder architecture of POD-DL-ROM with an LSTM autoencoder [9], in order to better take into account time evolution when dealing with nonlinear unsteady parametrized PDEs; second, it aims at performing extrapolation forward in time (compared to the training time window) of the PDE solution, for unseen values of the input parameters – a task often missed by traditional projection-based ROMs. Our final goal is to predict the PDE solution on a larger time domain $(T_{in}, T_{end})$ than the one, $(0, T)$, used for the ROM training – here $0 \leq T_{in} \leq T_{end}$ and $T_{end} > T$. To this aim, we train a POD-LSTM-ROM using $N_t$ time instances and approximate the solution up to $N_t + M$ time steps from the starting point, taking advantage of a LSTM architecture in a different form besides the POD-LSTM-ROM introduced before. These architectures mimic the behaviour of numerical solvers as they build predictions for future times based on the past.

## 2 Methods

Once discretized in space – e.g., through a finite element method – a nonlinear, time-dependent, parametrized PDE problem takes the form of a (high-dimensional) dynamical system:

$$\begin{cases} \mathbf{M}(\boldsymbol{\mu})\dot{\mathbf{u}}_h(t; \boldsymbol{\mu}) = \mathbf{f}(t, \mathbf{u}_h(t; \boldsymbol{\mu}); \boldsymbol{\mu}), & t \in (0, T), \\ \mathbf{u}_h(0; \boldsymbol{\mu}) = \mathbf{u}_0(\boldsymbol{\mu}). \end{cases} \tag{1}$$

In this context, denoting by $N_h$ the dimension of the FOM, $\mathbf{u}_h : [0, T) \times \mathcal{P} \to \mathbb{R}^{N_h}$ is the parametrized solution of (1), $\mathbf{u}_0 : \mathcal{P} \to \mathbb{R}^{N_h}$ is the initial datum, $\mathbf{f} : (0, T) \times \mathbb{R}^{N_h} \times \mathcal{P} \to \mathbb{R}^{N_h}$ is a (nonlinear) function, representing the system dynamics and $\mathbf{M}(\boldsymbol{\mu}) \in \mathbb{R}^{N_h \times N_h}$ is the mass matrix of this parametric FOM, assumed here to be a symmetric positive definite matrix.

In a parametrized context, the goal is to recover an efficient approximation of the solutions' set

$$\mathcal{S}_h = \{\mathbf{u}_h(t; \boldsymbol{\mu}) \mid t \in [0, T) , \ \boldsymbol{\mu} \in \mathcal{P} \subset \mathbb{R}^{n_\mu}\} \subset \mathbb{R}^{N_h}, \tag{2}$$

of problem (1) as a function of $(t; \boldsymbol{\mu})$ varying in $[0, T) \times \mathcal{P}$, called the *solution manifold*. DL-ROMs [3] and POD-DL-ROMs [2] describe both the solutions' manifold and the underlying reduced dynamics by means of a CAE and a DFNN. This step is completely data-driven and is based on an offline training of the networks on a set of full order snapshots, *i.e.*, solutions of the high fidelity model at particular instances of the parameters' space. In this way, the linear projection is avoided and a map solution $\rightarrow$ reduced manifold $\rightarrow$ solution is automatically derived from the data. Also, the evaluation stage is extremely fast, because it only requires to deal with a low dimensional regressor as it just considers the decoder structure of the architecture, thus allowing for faster than real-time computing.

The working scheme of POD-LSTM-ROM consists of several components:

(i) A (randomized) POD, or (r)POD [8], is performed on the matrix of snapshots used for the training, to reduce their dimensionality from the FOM dimension $N_h$ to the ROM dimension $N$.

$$\mathbf{u}_N(t; \boldsymbol{\mu}) = \mathbf{V}_N^T \mathbf{u}_h(t; \boldsymbol{\mu}), \tag{3}$$

being $\mathbf{V}_N \in \mathbb{R}^{N_h \times N}$ the POD projection matrix.

(ii) Snapshots are then sequentially stacked in matrices $\mathbf{V}_N^T \mathbf{u}_h([t_0 \ldots t_K]; \boldsymbol{\mu}) \in \mathbb{R}^{K \times N}$ ($K$ is the sequence length) which are then grouped in (mini) batches to allow the training.

(iii) These sequences are then fed to an LSTM encoder structure [4,15] (used during training only) to reduce the information coming from the input sequence into a lower dimensional manifold of dimension $n < N \ll N_h$:

$$\tilde{\mathbf{u}}_n(t_0; \boldsymbol{\mu}, \boldsymbol{\theta}_{enc}) = \boldsymbol{\lambda}_n^{enc}(\mathbf{u}_N([t_0 \ldots t_K]; \boldsymbol{\mu}, \boldsymbol{\theta}_{enc})). \tag{4}$$

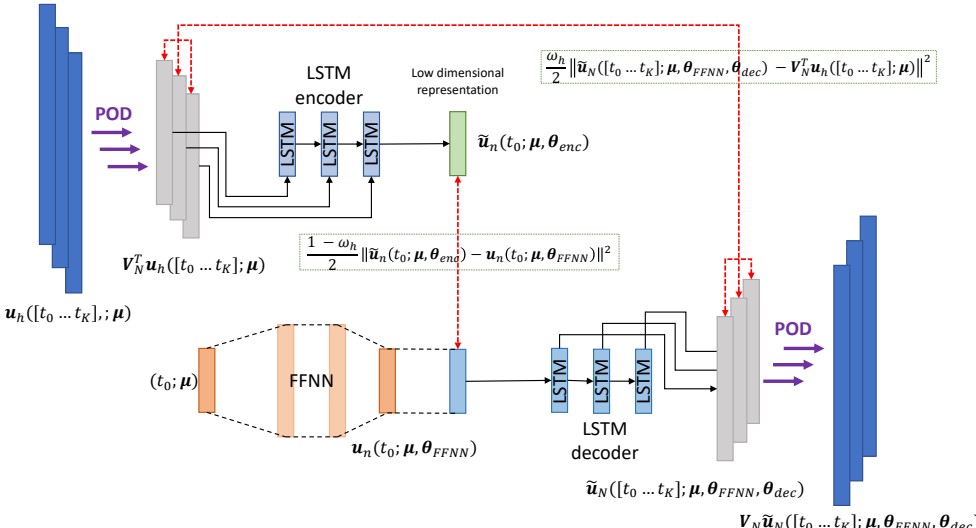

Figure 1: The POD-LSTM-ROM architecture.

(iv) A (deep) feedforward neural network acts as a regressor to predict the $n$-dimensional representation of the autoencoder starting from the parameters vector $\boldsymbol{\mu}$ and the starting time $t_0$:

$$\mathbf{u}_n(t_0; \boldsymbol{\mu}, \boldsymbol{\theta}_{FFNN}) = \phi_n^{FFNN}(t_0; \boldsymbol{\mu}, \boldsymbol{\theta}_{FFNN}). \tag{5}$$

(v) The decoder function of the LSTM autoencoder structure then builds the reduced nonlinear manifold upon the low dimensional representation inferred by the feedforward structure:

$$\tilde{\mathcal{S}}_N^n = \{\boldsymbol{\lambda}_N^{dec}(\mathbf{u}_n(t; \boldsymbol{\mu}, \boldsymbol{\theta}_{FFNN}); \boldsymbol{\theta}_{dec}) \mid \mathbf{u}_n([t \ldots t + K\Delta t]; \boldsymbol{\mu}, \boldsymbol{\theta}_{FFNN}) \in \mathbb{R}^n, \\ t \in [0, T - K\Delta t) \text{ and } \boldsymbol{\mu} \in \mathcal{P} \subset \mathbb{R}^{n_\mu}\} \subset \mathbb{R}^{N \times K}. \tag{6}$$

This structure contains the predicted sequences produced by the decoder, *i.e.* a $K$ time steps temporal evolution of the reduced order solution of the PDE problem with parameters $\boldsymbol{\mu}$, starting at time $t_0$:

$$\tilde{\mathbf{u}}_N([t_0 \ldots t_K]; \boldsymbol{\mu}, \boldsymbol{\theta}_{FFNN}, \boldsymbol{\theta}_{dec}) = \boldsymbol{\lambda}_N^{dec}(\mathbf{u}_n(t_0; \boldsymbol{\mu}, \boldsymbol{\theta}_{FFNN}); \boldsymbol{\theta}_{dec}). \tag{7}$$

(vi) These sequences are then expanded from dimension $N$ to the original full order dimension $N_h$ by means of the POD basis found in step (i), yielding the actual prediction of the PDE solution,

$$\tilde{\mathbf{u}}_h([t_0 \ldots t_K]; \boldsymbol{\mu}, \boldsymbol{\theta}_{FFNN}, \boldsymbol{\theta}_{dec}) = \mathbf{V}_N \tilde{\mathbf{u}}_N([t_0 \ldots t_K]; \boldsymbol{\mu}, \boldsymbol{\theta}_{FFNN}, \boldsymbol{\theta}_{dec}), \tag{8}$$

and finally a stack of $K$ full dimensional time sequential solutions is found.

(vii) To enhance the framework with better time extrapolation capabilities, the aforementioned structure can be equipped with an LSTM autoregressive model, trained on the same data. This additional paired network takes the last $p$ time steps predicted by the POD-LSTM-ROM and advances the ROM solution of $m$ forward time steps. Repeating iteratively this procedure, time extrapolation can be achieved.

Details on the architecture and the loss function can be found in Figure 1; we highlight that all the networks are trained simultaneously, combining in the loss function both the reconstruction error and the error on the low-dimensional representations of the FOM snapshots.

## 3   Results

We discuss here two test cases for the presented framework. The first one deals with an advection-diffusion-reaction (ADR) problem, in which some particles interact with a solvent by convection and diffusion, while reacting with it and degrading consequently. The second is an elastodynamics problem that considers a micro beam ($1mm \times 24\mu m \times 10\mu m$) fixed at both ends and subject to an internal force simulating a piezo-electric actuation. The FOM dimension is $N_h = 10657$ for the ADR case and $N_h = 7821$ for the elastodynamics case, respectively; the former problem has been

chosen to be equal to the one considered when testing the POD-DL-ROM framework in [2], while the micro beam one is described in [5]. ADR problem features a three-dimensional parameter space, with parameters influencing the degradation speed and the position of the particles source, while the elastodynamics problem depends on 2 parameters, influencing the amplitude and the frequency of the piezo-electric oscillating force. FOM data have been obtained by the framework implemented in [11]. Training has been carried out for the ADR case on 100 equally spaced instances of the parameter space, each yielding a 100 time steps FOM solution in the time interval $[0, 10\pi)s$. Instead, 108 instances of the parameters space and a 60 time steps temporal evolution in $[0, 14.85)s$ have been considered for the elastodynamics one. The sequence length for the LSTM autoencoder has been chosen to be $K = 20$ for both cases, as it provided a good accuracy/prediction speed trade-off.

A selection of simulation results is reported in Figure 2. Approximation at testing stage is very good (mean rel. error: $O(10^{-4})$ for ADR and $O(10^{-6})$ for the microbeam) and makes the method reliable for multiple input-output queries situations, as it provides high accuracy when generalizing on unseen parameters instances. Furthermore, the remarkable accuracy remains stable across the entire time domain, even when extrapolating, as shown in Figure 3. Some indicators for the reduced order vectors POD-LSTM-ROM inference time to predict the entire test sets are $t_{NN}^{min} = 0.0807s$, $t_{NN}^{avg} = 0.0892s$, $t_{NN}^{max} = 0.3202s$ for the ADR problem and $t_{NN}^{min} = 0.1057s$, $t_{NN}^{avg} = 0.1143s$, $t_{NN}^{max} = 0.1477s$ for the elastodynamic case (they have been obtained on 100 different runs of the prediction framework). Note that the computations (run in local on a Intel Core i9, 16 GB RAM, NVIDIA GTX1650 machine) allow for a faster than real-time simulation, particularly useful in many fields of application where timing is critical.

Therefore, the proposed method shows extremely good generalization capabilities for what concerns physical parameters. Extrapolation in time when relying on the procedure described at point (vii) in Section 2 also provides reliable results, even on extrapolation windows of length up to 16 times the training window. A still open question concerns the accuracy of results in the case of highly nonperiodic or chaotic scenarios, for which additional investigations are ongoing.

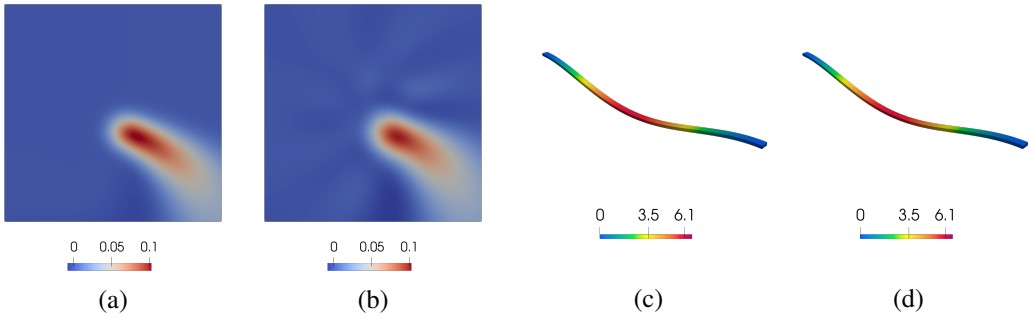

Figure 2: Simulation results for the ADR case at time $3.9\pi s$ (FOM (a) and POD-LSTM-ROM (b) results) and for the elastodynamics problem at time $15s$ (FOM (c) and POD-LSTM-ROM (d) results).

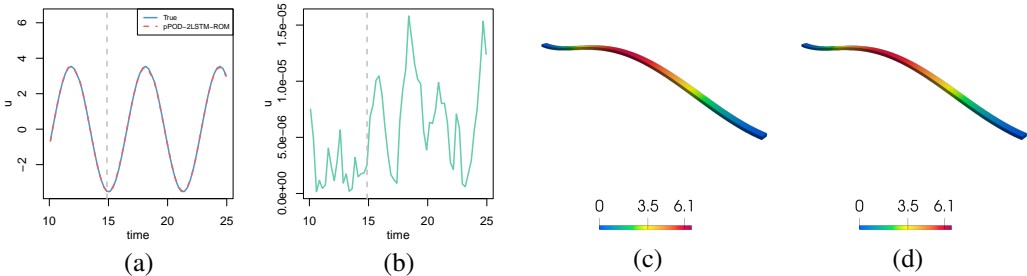

Figure 3: Time extrapolation results for the elastodynamics problem: time evolution for a single degree of freedom (a) and relative error (b); the dotted line represents the extrapolation starting time. FOM (c) and POD-LSTM-ROM (d) results at time $t = 24.5s$.

# 4 Conclusions and future work

In this work, we presented a deep learning-based framework that allows fast approximation of PDE systems solution. We obtained faster than real-time simulations that are able to preserve a remarkable accuracy across the entire time domain considered during training. An extension of the method enriching it with time extrapolation capabilities has also been presented and tested.

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
