# OpenReview forum: "Long-time prediction of nonlinear parametrized dynamical systems by deep learning-based ROMs"
_NeurIPS.cc/2021/Workshop/DLDE — DLDE Workshop -- NeurIPS 2021 Poster_

### Official Review · Reviewer_RX7E · 2021-10-02
**Nice paper with a promising direction of research, a couple of clarifications needed**

**Confidence:** 3

**Review:**

This work improves deep learning-based reduced order models for solving PDEs by using LSTMs in the encoder and decoder, taking advantage of their ability to work with time series data. The paper makes good technical contributions to make training less expensive (by reducing the dimensionality of the problem) and improve the extrapolation ability of the model (by using LSTMs in the encoder/decoder).

I have a few minor points to clarify:
- It is not immediately clear how the model is used at test time. Is the LSTM encoder used? Or just the feed forward network?
- The diagram is difficult to decipher, could this be made neater?
- The results section does not compare the model against DL-ROMs or POD-DL-ROMs, is it possible to include either of these benchmarks instead of simply listing where to find them?

Typos:
- Line 118 should say "as shown in Figure 3" instead of "as shown in 3"


**Score:**

3: Good paper

---

### Official Review · Reviewer_uauU · 2021-10-08
**Interesting work and well written; more testing and comparison with other methods would be beneficial**

**Confidence:** 3

**Review:**

This work improves deep learning-based reduced order models for solving PDEs by using a double reduction through randomized POD and LSTM encoder. The LSTM AE is used to predict future time steps. The paper makes good technical contributions to make training less expensive and improve the extrapolation ability.

However, there are some improvements I would suggest for the paper:
(1) Figure 1 is helpful but more explanation on the architecture is required. For instance, was there any study on which POD helps obtain better results, and what is the structure of the LSTM auto-regressive model used to make predictions into the future time steps.
(2) How many parameters does this architecture require and how does it compare in that regard to previous work?
(3) There needs to be a comparison of numerical results in the Results section with other methods cited by the authors.
(4) The same comparison should be done regarding the training stage to show the improvement of the current model with respect to previous work.

I understand that this is fundamentally a data driven approach, but I wonder on the practical usefulness in cases where no FOM solutions are available for training. In addition, in cases where data is available but one requires a solution at a time somewhere in between two consecutive time steps, can this architecture be of any help? And, at last but not least, it would be useful to be able to somehow asses the error of the extrapolation (prediction) when there is no data to compare to. Would it be possible to give a confidence interval of values to the user instead of only one prediction?

**Score:**

3: Good paper

---

### Official Review · Reviewer_849F · 2021-10-12
**Clear framing and approach with limited results but potential for an experimental paper**

**Confidence:** 4

**Review:**

Long-time prediction of nonlinear parametrized dynamical systems by deep learning-based ROMs

Summary:
The paper proposes a learning-based surrogate of time-dependent parametrized PDEs. The novelty lies in replacing the neural network (CAE/FCNN) of a previous work (POD-DL-ROMs) with an LSTM. In summary, the paper provides a solid framing and clear description of the approach with limited results to support the abstract claims, but potential for an experimental full-length paper if literature review and experiments are extended.

Review:
Pro:
- The mathematical description of the approach is very clear, ordered, complete, and concise.
- The literature review correctly categorizes the proposed method in the traditional ROM literature without purely focusing on learning-based methods.
- The results show promise
- The proposed model architecture resembles Fourier neural operators (FNO). While FNO uses a Fourier transform to preselect good prediction features, this POD-LSTM-ROM paper is using POD. It remained unclear whether POD-LSTM-ROM is losing accuracy through projecting the solution onto the POD space. It would be very interesting to 1) analyze the effect of introducing a skip-connection similar to FNO, 2) argue which functions can and cannot be approximated with the current POD-LSTM-ROM architecture.

Improvements:
- Most importantly, the paper's main argument that the method can extrapolate to new parameters and time-steps remained unproven/-shown. First, the authors have not described which parameters, mu, have been used to generate their testing dataset. This makes it very challening to evaluate whether the proposed approach shows potential for extrapolating parameters [line 116-117]. The approach does evaluate the temporal extrapolation capabilities, however, presumably only one spatial point is shown to evaluate the approximated ADR (Fig 3). It would be great if the authors could clarify whether Fig 3a, 3b is only one spatial point or a spatial average and how it has been chosen. Alternatively, the authors could also improve the paper by adding a plot with 1 spatial dimension on the y axis, time on the x-axis and the error on the z-axis (colored).
- The abstract argues to reduce the number of necessary training samples, but the argument is not adressed in the main body of the paper. I would recommed to either remove the argument for reducing training samples from the abstract or adding a section that argues for reduced training samples.
- The paper seems to have limited theoretical novelty as it "only" replaces an FCNN or CAE with an LSTM. That does not seem to bad for a full-lenght paper if the paper convinces through an extensive experiment section that compares the method to other learning and non-learning-based ROMs in terms of training, runtime complexity, and accuracy.
- The title promises a method for "long-time" prediction. However it seems unclear whether the predictive error will accumulate over long-time prediction. It would be great to add a plot that shows the time until the method diverges or compares the error doubling time of various methods.
- It would be very helpful to clarify in which exact situations the proposed learning-based ROM is useful. For example, in the case when limited simulation runs exist, the underlying PDE too high-dimensional for traditional methods, and we are interested in interpolating over parameters, mu.

Minor:
- The following sentences need a reference: 14-18, (POD-Galerkin ROMs) 19, 52-53,
- Add arguments on why ADR and FOM have been chosen as sample equation
- The world "real-time" could be defined [l. 123]
- The abstract uses the word multi-fidelity, but it remained unclear what is meant by multi-fidelity. I would recommend to remove this word or define it in the text. Especially, as this could be mistaken with multiscale PDE solvers.
- It would be helpful to define or reference the definition of a reduced or nonlinear trial manifold
- FFNN could be defined in l.80
- It would be easier to read the training set-up if the set-up is first described for the one equation and then for the other.
- The method is stated to approximate the ADR "very good" (l.116) without providing an error value.
- I am a bit unclear whether (iii) is used during training and testing or only training. I think only during training, but it would be helpful to add a sentence on this.

Thank you for the read!

**Score:**

3: Good paper

---

### Decision · Program_Chairs · 2021-10-16

**Decision:**

Accept (Poster)

**Comment:**

Reviewers agreed on paper acceptance.